# Modeling the formation of a worldwide health network fighting TBC: Key drivers in policy, management and governance in developing countries and global health institutions

**Mauro Gallegati[1], Milena Lopreite [2]\*, Michelangelo Puliga[3]**

**1** Department of Management, Marche Polytechnic University, Ancona, Italy, **2** Department of Economics, Statistics and Finance G. Anania, University of Calabria, Arvacata of Rende, Italy, **3** Department of Economics, Cà Foscari University, Venice, Italy

\* milena.lopreite@unical.it

**Data availability statement:** All relevant data are within the manuscript and its Supporting information files.

## Abstract

We present an original study on the usage of a model of network formation to analyze the X (formerly Twitter) friendship network relative to the health organizations that are fighting a specific infectious disease such as Tuberculosis (TBC) as well as how the network evolves over time. Using this model, that mixes information from social media and the underlying reconstructed economic network of their actors, together with disease incidence information, we can then study how the TBC global health network (GHNs) works. Specifically, we investigate the key drivers of this global network partnerships as well as the interplay between economic, social media, and disease incidence. The network diversity (a measure of node size dispersion), has been identified as the leading feature for the network growth, while improving its resilience. We use these insights to suggest better health strategies especially targeted to weak GHNs operating in low and middle income countries that often lack funding, coordination and the capability to attract new donors.

## Introduction

Non governmental organizations (NGOs) partnerships are a growing trend of modern times taking the form of consortia promoting global aid programs and humanitarian actions, especially in low and middle income countries (LMICs) that globally represent the regions with the poorest health conditions [9–11]. These partnerships have been recognised as a "powerful tool" to reduce global disparities, advancing the goals of aid, justice and promote health equity [10]. However, managing these activities requires a good "governance" in terms of coordination, decision-making about the targets, priorities and allocation of resources and the ability to secure adequate funds for each research program in such a way to reduce the proliferation of donor-recipient redundancy, duplication, and administrative burden that may lead to less cohesion and in turn inefficiencies the partnership structure [5].

**Funding:** The author(s) received no specific funding for this work.

**Competing interests:** The authors have declared that no competing interests exist.

In this context, a crucial role for a good health system management and effective health policy interventions is depicted by those organizations that fight specific diseases, such as Malaria, Tuberculosis (TBC) or Pneumonia. Their activities require a transnational effort in terms of health vigilance, safety, governance, financing, service delivery, human resources, information technology, and many other local activities directed to improve the clinical conditions of patients (vaccinations, drugs, diagnostics) [18].

Specifically, the TBC disease is playing a pivotal role in antibiotic resistance [1,6], it is among the most important co-morbidities for HIV patients [2,7] and it is having a strong impact on general population health conditions, and on their long term possibility to be part of an active and healthy workforce [3,8].

Nowadays, the NGOs are using X ( the former Twitter platform) or Facebook to enact social media campaigns addressed to improve aid funding effectiveness, diffusing awareness, attracting attention and involving donors [4]. Along the years there has been an increase of the NGOs and other international organizations that have official accounts on the X/Facebook platform and they are connected to each other with a friendship relationship.

The X/Facebook friendship network may be relevant for coordinated actions and trust development over a long-term partnership in the domain of disease-fighting, with the goals of improving governance to promote health equity, design better funding policies to empower low and middle income countries' health systems, and supporting better medical interventions in these countries. However, by analyzing the partnership on X platform, it emerges that several networks may be struggling while others flourish and this aspect may change after a disaster or an epidemic. For example, before COVID-19 pandemic the Pneumonia disease despite its incidence , especially in the pediatric age, was mostly a neglected disease, with just some local organizations having a dedicated mission, while after the COVID-19 outbreak the consortia fighting this disease received a renewed attention from governments and policy-makers. In this context, the tools of social network analysis, combined with those of network science and statistical physics may help in detecting the presence of poorly connected communities (i.e pneumonia network), as isolated nodes that are not forming connections indicate a weak coordination, finally central nodes that are acting as hubs in an efficient network are crucial in fighting a specific disease (in the case of tropical diseases the hubs are in most cases the international official organizations such as WHO or vaccine alliances such as GAVI or international programs like StopTB). We also point out studies that have clearly shown the importance of defining the internal dynamics of social networks, defining key players and the effects of isolating / promoting their behavior with appropriate coordinated actions [23].

Although there are many analyses that underline several advantages of the organizations' partnership in terms of improving governance and efficiency [18,19] there are still few studies on network discovery that try to both identify and measure the network components using the methods of social network analysis: a tentative was performed by [20] that used manual screening of websites, identifying only a handful of global health network organizations (approximately 208 organizations). On the other hand, instead, the literature on the network formation models, and also on agent based networks is particularly rich in the field of epidemiology where this category of algorithms can promote population-level inference from explicitly programmed micro-level rules in simulated populations over time and space [12]. Estimating the disease spread starting from social interactions is particularly interesting, as it allows to evaluate the diffusion speed of the infectious disease, and at the same time the effectiveness of a lockdown policy like in the work of [14]. Against this background we contribute with respect to the existing literature as follows: a) on the side of network science we employ a model of network formation to simulate the effects of node diversity (measure of dispersion in node economic size and popularity distributions) and of a network with more donors ; b)

on the side of the epidemiology field we analyze the impact on the network formation of the increase of disease's levels.

Thus, we can summarize the contribution of our analysis in four steps: Firstly, starting from the work of [5] we refined the X (formerly Twitter) friendship network of users involved in the TBC fight by adding to the algorithm a new layer of classification made according to a manually annotated database of these users that helped to discriminate them in 4 different account types: institutions (such as UN agencies, or official centers for disease prevention i.e. CDC in the US), donors (i.e. charities, or companies that - according to their description - are active fund donors in the TBC fight and usually live in rich countries), receivers (Non Governmental organizations or charities that - sitting in poor countries - collect funds for their action) and others (such as experts, doctors). Specifically, the ML Random Forest classifier analysis has been hyper-tuned on the database of these four users' classes (institutions, receivers, donors, and others) that was also used to train the model with cross-validation, reaching an average F1 score of 0.94.

The ML model was trained using the following features to classify each node:

Binary Country Location: A categorical variable distinguishing between developed and developing countries.

Internet Domain: The top-level domain of the node (e.g., .gov, .int, .edu, .com, .org), where .gov and .int domains, along with direct references to UN agencies, clearly indicate institutional entities.

Twitter User Description: This was treated as a text corpus, with standard TF-IDF (Term Frequency-Inverse Document Frequency) applied to extract features. Descriptions explicitly identifying users as institutions, charities, or individuals (e.g., doctor, expert) were particularly valuable.

During pre-processing, one feature has been derived from web domain names using regular expressions to extract domain suffixes (e.g., .gov, .eu, .com). We also created a feature indicating the presence of references to UN agencies (e.g., "unhcr") or local WHO agencies within the text. Standard text processing techniques for English were applied, including stop-word removal, limiting n-grams to two words, and setting a maximum of 200 TF-IDF features. A RandomForest classifier was used, with simple hyperparameter optimization performed via grid search for the number of estimators and maximum leaf depth. The classifier operated in "balanced mode," applying weights to classes during the training phase to address potential class imbalances.

Then, with this new information we can sketch a model of network formation, and use it to examine the interplay among the different users' roles.

Secondly, we investigate the *economic determinants* of the NGOs friendship network formation focused on TBC and its evolution over time. For this goal we use, as a proxy of the NGOs economic node size, the overall financial flows that these nodes have exchanged over the years in TBC-related projects as reported by the specialized platform `d-portal.org`. Filtering all information on these exchanges we build a network based only on supposed "economic node size" as sum of total financial flow, and financial flow transactions (expressing the links). This second network shows a partial overlap (around 20 – 40% depending on the strictness of the matching criteria) with the network that we built before using exclusively the "X friendship". Importantly, the d-portal network (details of the network can be found in Table 1) reports more information on institutional actors (i.e there are nodes that are not present in the X network and that are mostly ministries and governments).

Although the two networks just discussed above are not fully equivalent and overlapping, and they have also a different domain (X is a social network, and the other (d-portal network) is a collaboration and funding network), without losing generality we can use information

drawn from both networks to reconstruct the X network with a unique model of network formation. In simple words we use and merge the information of the two networks to establish how the X friendship network is likely to be formed.

This is a fundamental point of our research: the new network formation model is aimed at reconstructing the topology of the X network by adding the information drawn from a proxy network (d-portal) on the economic size of the nodes. Methodologically, preferring the reconstruction of the popularity network on X has several reasons: a) X is a public network and everyone can check friendship relations between users. D-portal, instead, tracks financial flows and it does not have a direct link concept: for example, if two institutions have not exchanged funds but they are part of the same project this statement it is not displayed in the d-portal records. Thus, despite being part of a project, they appear erroneously disconnected in the financial flow network; b) Popularity on X is directly measured by the number of followers, conversely, economic size is only inferred from d-portal via proxy data on financial transactions and it may be incomplete. Therefore, strategically, we use d-portal only to guess that the economic distribution of the node sizes is reasonably skewed and modeled close enough to the correct orders of magnitude of the true but unknown economic size of the organizations. The sum of the financial flows per node in d-portal database ranges from 10k USD to a maximum of 10B USD and we accept these values as the correct order of magnitude of the true unknown economic sizes.

Thirdly, we simulate in our network formation model the linking of the nodes according to three conditions: a) popularity on X; b) economic attractiveness as expressed in the d-portal database; c) intensity of the disease (drawn from the World health organization TBC incidence pages) https://www.arphs.health.nz/public-health-topics/immunisation/countries-with-a-high-incidence-of-tb/. Then, using a classification algorithm, we distinguish the nodes into four categories: donors, receivers, institutions and, others. Our aim is to employ this model to gain useful insights on the X friendship network formation and its possible evolution over time, on how changes in disease incidence, or a change of donor fraction can alter the network formation.

In sum, the usage of the model of network formation together to the nodes' size diversity (expressed as how much are dispersed the nodes' sizes in terms of economic and popularity) can help to understand how an example of global health networks works (in our case the TBC network, but other GHN should follow the same rules), to investigate the prevailing reasons for node linkage and, in the context of policy making and health management, suggesting best practices for network expansion, improving governance policy and in turn promoting health equity.

## Methods

We start our investigation from the work of [5] focusing on the X (formerly Twitter) *friendship* network of users involving in TBC programs. In the X platform the "friendship network" is the dual of the "followership network" (opposite link direction). By definition in the X platform the users that a user follows are called "friends", while the users that are following him/her are called "followers". The difference is crucial: a user can only have a limited number of friends, instead, can be followed by a large number of followers (usually, stars, and famous people have a lot of followers and very few friends)- in addition, only the updates of friends are visible to a user. Followers are, in fact, voiceless. This asymmetry plays a fundamental role in our research: only the friendship network is relevant for information diffusion, and for possible endorsement and collaboration. To build the entire network of X friends we started from a seed of recognized users that are the organizations fighting TBC (such as

"endtbnow"). Specifically, in the starting paper of our research [5] has been implemented a Machine Learning (ML) classifier (see footnote for details) that is able to discriminate among relevant/non-relevant users: in this way a large network of 1838 nodes, and 7441 links representing organizations or professionals fighting Tuberculosis has been recovered. The ML model for user classification has been trained on the following features. Binary country location (i.e. developed vs developing country). Internet domain of the node (.gov, and .int are clearly referring to institutions, together with references to UN agencies) Twitter user description: the description clearly reports if a user is an institution or a charity, or an individual (doctor/expert). The user description is treated as a corpus and the ordinary TfIDF treatment applied. Regarding the pre-processing, one feature is obtained by regular expression elaboration of web domain names to get domain suffixes (.gov, .eu, .com etc.) , and the presence of reference to UN agencies (i.e. unhcr ) or WHO local agencies.

Classical text processing for English is applied (stop words removal, ngram limited to 2 words, and tdidf max features = 200). The classifier is a RandomForest and a simple hyperparameter optimization has been obtained with grid search (for number of estimators, and max leaf depth). The classifier is in "balanced mode", i.e. it applies weights to classes during the training phase.

At the same time an equivalent network of organizations involved in TBC-related programs it has been built using the data drawn from *d-portal.org*: a specialized platform that tracks and monitors the financial flows for non governmental organizations, governments and institutions filtering them by topic (i.e. projects related to healthcare, infectious diseases such as Tuberculosis, Malaria etc.)

Specifically, since the small charities and NGOs have no strict requirements in terms of publishing their financial statements and, there is no direct way to estimate their assets, in order to infer the economic size to these nodes (NGOs, charities) we used as a proxy the measure of the total flow of money that each organization managed for TBC-related projects as reported by d-portal database. Finally, the links in this economic network are the result of participating in common projects (i.e. two organizations are connected if they are involved in one or more projects). For a better understanding a quick comparison of the two networks characteristics (economic and social oriented) is shown in Table 1.

## Modeling network formation

Before to introduce in more detail our analysis, two relevant starting points are necessary to better understand the model of network formation: a) the analysis of the nodes' size dispersion characterized by few large nodes mixed with many small ones in X and in the d-portal database (respectively for popularity and economic size) leads to a skewed distribution for the both measures (number of followers and incoming financial transactions as a proxy of economic size). The results are shown in Fig 1 left and right; b) we assume, as a working hypothesis, that an exponential distribution is enough to model both distributions: this distribution (see Eq 1) has the advantage of having a single parameter $\alpha$ and being extremely easy to interpret. In addition to that, this distribution can be fitted with sufficient accuracy to the popularity and economic size distribution (see Fig 1 left and right) at least in the 5th-95h percentile interval of data.

In this study, starting from the original X network of TBC [5], we introduced a new ML classifier to identify four different nodes' account types: a) official institutions such as WHO, or CDC; b) donors like charities in developed countries; c) receivers as charities in poor countries; d) others actors like experts, doctors and scientists. The classes within our dataset are significantly unbalanced as shown by the values predicted by the Machine Learning classifier

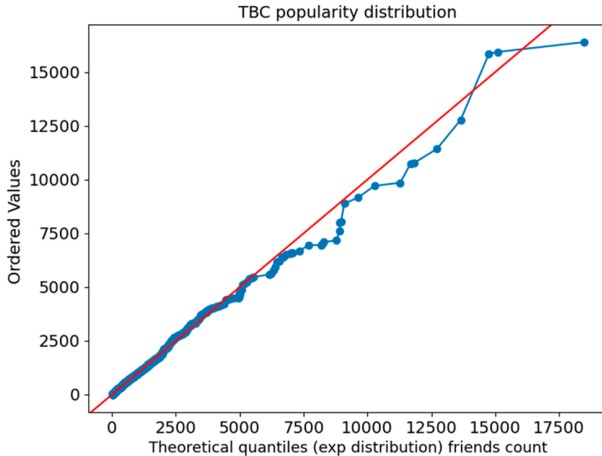
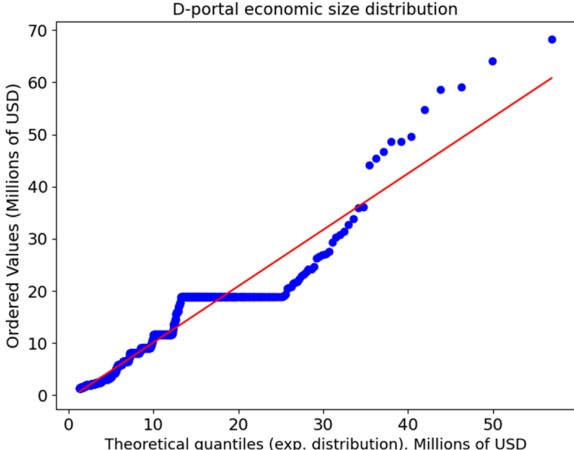

**Fig 1. Left side: Q-Qplot of the distribution of node popularity (exponential).** Right side: The distribution of the Economic size of NGOs in D-portal in Millions of USD fitted with a theoretical exponential distribution (in the interval 5-95th quantile, excluding the largest NGOs that moreover are institutions, rather than no profit organizations

from the original TBC network model [5]: Donors 1368, Institutions 88, Others 93, Receiver 289. Notably, the potential donors are approximately 77% of the total number of nodes in the dataset. We employed a standard 80/20 training/test split, performing 10-fold cross-validation (CV=10). Model performance, and F1 score has been averaged across these folds. Despite this class imbalance, the Random Forest model achieved an averaged F1 score of 0.94. This score was obtained after hyperparameter tuning and by applying "balanced" weights for class re-balancing during training. For a complete analysis, we provide the confusion matrix , which compares true and predicted classes, in the S1 Fig of the Supplementary Materials.

The set of X users and their relative description was first manually classified in donors, receivers, institutions, and others to create the training set, then a ML model (Random Forest) has been hyper-tuned on the training set to reach an accuracy of 0.8 and to generalize its classification in the entire dataset.

Once the network of TBC has been further classified, we introduce the algorithm detailed below to model the network formation following several custom rules that describe the combinations of the nodes and the mechanism of the link formation. In our model receivers and donors will have a prevailing economic interest in forming ties, instead institutions and others will not necessarily follow economic reasons to connect each others. We assume that the nodes will connect as follows:

1. Donors and Receivers will form a link based on disease level incidence $d$, economic interest $e$ and social media popularity $p$
2. Donors and Donors will form a link according to social media popularity (in this case the disease level is irrelevant as they usually sit in countries with no TBC outbreak)
3. Donors and Institutions will form a link according to social media popularity and disease incidence.
4. Donors and Others (i.e. no receivers) will form a link according to social media popularity and disease incidence.
5. Receivers and Receivers will form a link according to disease level and social media popularity (they compete in the same attention space).

6. Receivers and Institutions will form a link according to disease level, economic interest and social media popularity
7. Institutions and Institutions will form a link according to disease level and social media popularity.

The rules of the links' attachment are commutative and it holds the assumption that the likelihood of link formation does not depend from which node is starting the connection (for instance, a connection donor to receiver is equivalent to that of receiver to donor).

In the network formation model every link is formed with a probability that follows respectively a negative exponential law ( see Eq 1) for economic interest and social media popularity and a random uniform distribution for the disease intensity.

In the Eq 1) alpha is the only parameter of the distribution, instead $p(x)$ is the probability of $x$ a variable that represents the economic size distribution or the popularity distribution. From a mixture of these two distributions (economic and popularity sizes), according to the rules above - mapped with the function described in Eq 2 - the reconstructed network will form with a *node degree distribution*, and *local clustering distribution* that is similar to the one of the original X network [5].

$$p(x) = \alpha e^{-\alpha x}, \quad x \in (0, \infty) \tag{1}$$

Running a Monte Carlo simulation of the link generation that follows the distributions (exponential, uniform) just discussed above we can establish the presence of links' connections according to the Eq 2: whether exists a random number $R$ in the interval 0–1 that is lower than the combined effect of the probability $P$ of all factors we can affirm that there is links connections. In particular, node $A$ and node $B$ with respectively economic size $e_A$, $e_B$, popularity $p_A$ and $p_B$, sitting in countries with disease levels $d_A$ and $d_B$ will form a link if and only if it holds the relationship shown in Eq 2:

$$R \leq P \begin{cases} max\left(\frac{e_A+e_B}{2}, \frac{p_A+p_B}{2}\right) \cdot max(d_A, d_B) & \text{if A,B are donors/receivers,} \\ max\left(\frac{e_A+e_B}{2}, \frac{p_A+p_B}{2}\right) \cdot max(d_A, d_B) & \text{if A,B are institutions/receivers,} \\ \frac{p_A+p_B}{2} \cdot max(d_A, d_B) & \text{if A,B are others/receivers,} \\ \frac{p_A+p_B}{2} \cdot max(d_A, d_B) & \text{if A,B are both not receivers,} \end{cases} \tag{2}$$

where the probability $P$ expresses the node attachment during the Monte Carlo simulations. The economic interest becomes important only in the case of mixed nodes with one being a receiver(receivers vs donors or receivers vs institutions). Moreover, if the nodes are mixed and they have the same popularity, and comparable disease level (in their countries) the economic interest will prevail on the other measures (popularity, disease level).

Thus, with the rules just mentioned, we ran several experiments (Experiment 1,2,3) relative to the network formation varying the proportions of donors over the total, the economic size and, popularity distributions shape and the incidence of disease intensity.

Our model uses an exponential distribution for the data in the both case: node popularity in the X platform (expressed as number of followers) and the economic size in d-portal (inferred from the financial transactions registered in the d-portal platform). If we look at the interval of variation (difference between 90th and 10th percentile of the data distribution), we notice that the ratio $q90/q10$ is clearly more pronounced for the economic distribution, rather than for the popularity on X (Table 1, column "Value"). To assess the correctness of the network formation model a simple test was performed to reproduce the node degree distribution

obtained in the original work of [5]. Specifically, in our experiment the node size attributes (economic and popularity) will determine the linking distribution (node degree): by changing the parameters of our model relative to economic/popularity distributions and using an extensive Monte Carlo simulation we can then find the parameters' range that allows to form a network and it has a compatible degree distribution with the original graph from [5].

Specifically, our simulation and analysis follows three steps:

Firstly, we vary the economic node size $e$ and the X popularity $p$ according to two different Exponential distributions with coefficients $\alpha_p$ and $\alpha_e$. Taking into account the ratio $\lambda = \alpha_p/\alpha_e \in (0.01, 100)$ that it is a sufficiently large interval we can then study the cases when economics or popularity are respectively prevailing. To understand the effects of the ratio $\lambda$ we need to remember that the mean of an Exponential distribution is given by :

$$E = \frac{1}{\alpha} \tag{3}$$

then the smaller the exponent $\alpha$ the broader the distribution is. Practically, when $\lambda > 1$ is the economic size that prevails as $\alpha_e < \alpha_p$, when $\lambda = 1$ there is equilibrium, and when $\lambda < 1$ the popularity dominates. In this experiment the disease level is constant and unitary. To be noticed that - according to the rules we established - the economic size is determinant only for receivers, if the nodes that are randomly selected in the algorithm to form a link are not receivers, but for instance they are two institutions or a donor and an institution, their linkage probability is supposed to be proportional only to node popularity and/or disease level (Eq 2).

Secondly, we use the same experiment above this time by changing the disease level to capture the effects of the disease intensity in the node formation. The disease will vary according to a linear trend in a range (0,2) of arbitrary units (Fig 3).

Thirdly, while repeating the experiment with the same intervals of $\alpha_e$ and $\alpha_p$ we change the donors' fraction of the network in the large interval (5,20) per cent. Adding more donors to the network will increase the possibility for receivers to get funded (Fig 4).

In conclusion, in all experiments when a link is formed we investigate if the link has been formed by a prevailing economic node size $e$ or popularity $p$, or by "other reason" simply counting during the simulation process how many links over the total are formed by the economic, popularity or disease reasons (Eq 2). In this way we can investigate the prevailing causes for link formation while checking that the formed network has a distribution of node degree that is similar to the original network shown in [5] (see Fig 2).

Understanding the reasons of link formation may be useful to better address the health interventions while improving the partnership, governance for a more resilient Global health network especially in low and middle income countries.

## Results

This section presents the results of three distinct experiments (Experiment 1, Experiment 2, Experiment 3) on network formation. Our primary goal was to reconstruct a network where node connections are driven by a formation model considering three key indicators for each node: economic size, popularity, and disease level.

Overall, the experiments reveal that the model's performance in link formation is significantly influenced by the interplay between popularity and economic interest, and much less by the incidence of disease. Specifically, popularity plays a more crucial role when the distribution of economic node size (governed by its exponent $\alpha_e$) is less impactful (Fig 3-left part). Conversely, when economic diversity is more pronounced, economic interest becomes the

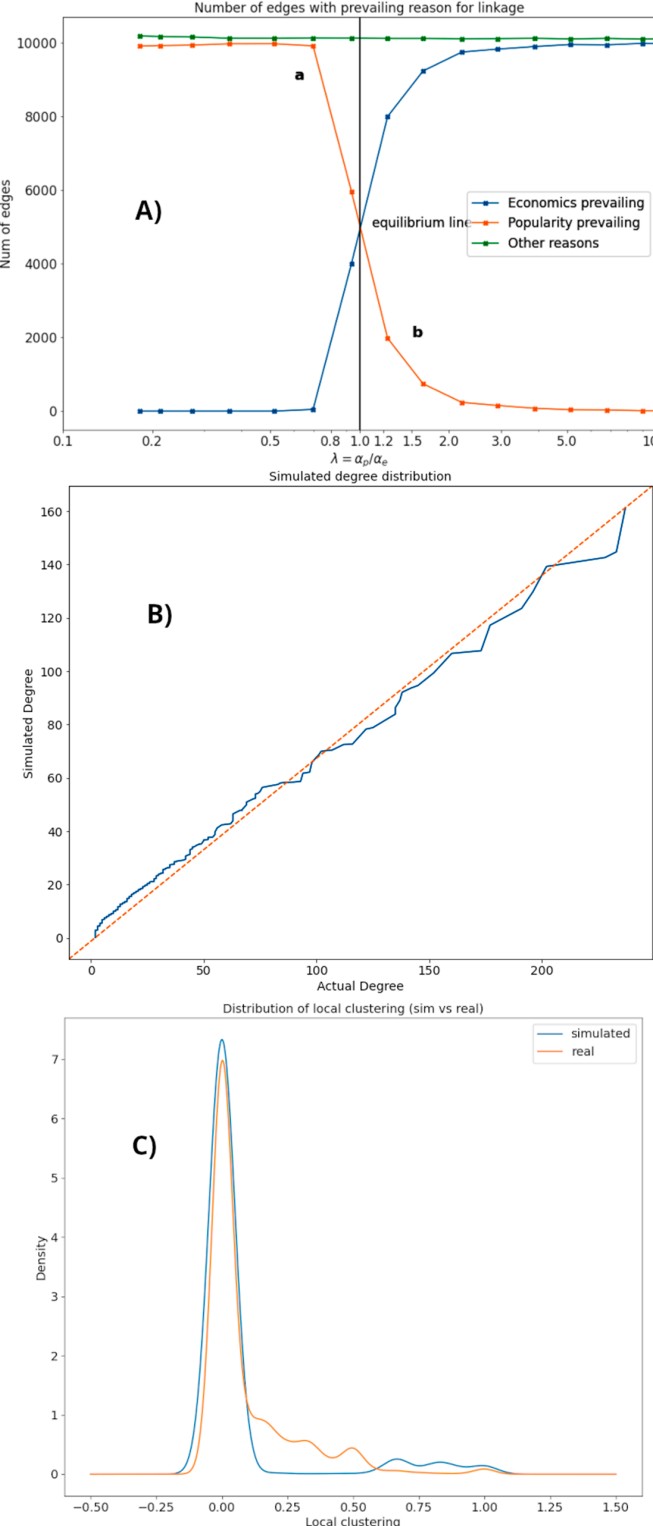

**Fig 2. Top A: Impact of varying economic and popularity size distributions on network formation.** By adjusting these parameters, we can observe how different levels of economic influence or social popularity drive the creation of links within the network. Middle B: Reconstructed degree distribution (quantile plot) generated by the simulation, representing a scenario where economic size and popularity exert balanced influence on node attachments. Bottom C: Comparison of the local clustering distributions for the real and simulated network, providing insight into how well the model replicates the density of connections around individual nodes.

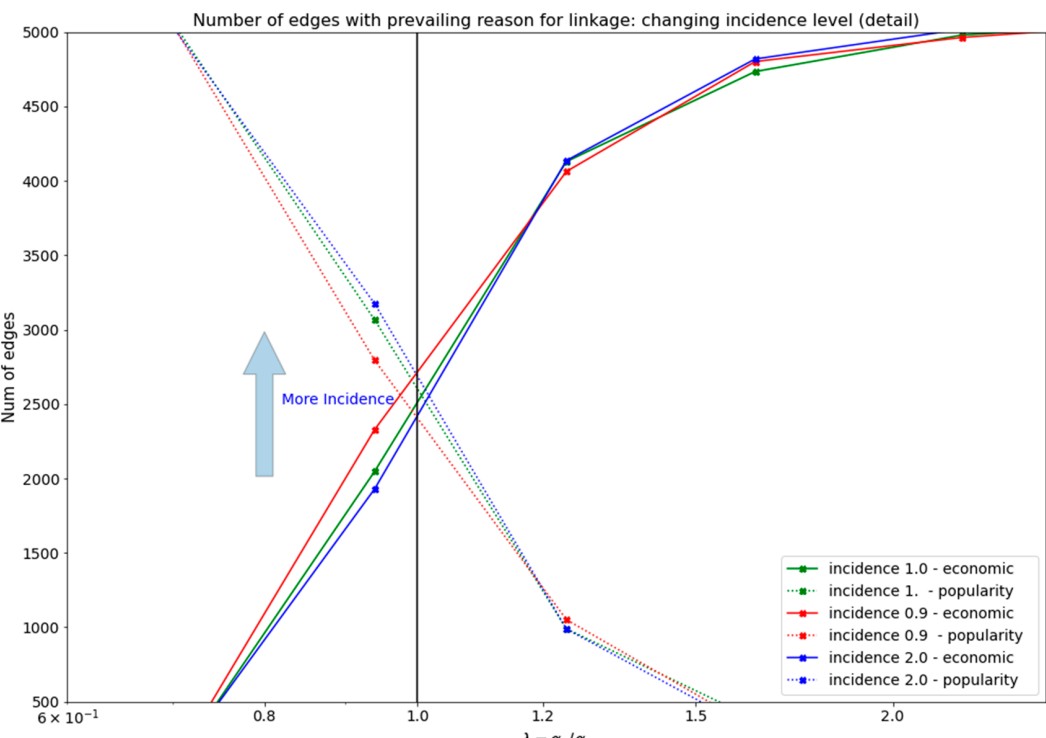

**Fig 3. In the left side is shown the effect of a change in the disease level (arbitrary units) on the link formation process due to economic size and popularity measures.** The picture detail displays how the graphs based on economic vs popularity reasons shift according to the increased incidence level.

dominant factor driving linkages. Detailed findings for each experiment are presented in the following sections.

### Experiment 1

In the first experiment, we simulate the link formation when the node sizes are measured by economic size and popularity and are distributed according to two different exponential distributions, given a fixed disease level incidence. Varying the ratio of the two exponential functions' exponents starting from the Eq 1 we obtain a link formation measure - expressed as number of links created following an attachment process - that is shown in Fig 2A). In this simulation by changing the ratio $\lambda$ (that it is the ratio of the two exponents of exponential distributions represented by Eq 1), we establish which indicator of the node size's dispersion (economic size or popularity) prevails in the link formation process. By counting how many links are formed by economic size or by popularity reason, we find two different regimes (region $a$ and region $b$). When $\lambda = \alpha_p/\alpha_e = E_e/E_p < 1$ the link formation is mostly due to nodes that connect each others if one of them or both are have a significant popularity (in this case the average popularity $E_p = 1/\alpha_p$ is higher than the corresponding $E_e = 1/\alpha_e$ (region $a$). If $\lambda = \alpha_p/\alpha_e = E_e/E_p > 1$ indeed the prevailing linkage is due to economic reasons (region $b$).

Interestingly, in the plot mentioned above the regions indicated by $a$ and $b$ where the network linkages is based on popularity and economic size are not perfectly symmetric. Instead, if the nodes' interplay between economics and popularity were equivalent the curves should

be perfectly symmetric respect to the vertical line placed on $\lambda = 1$. The reason for this divergence can be found in the details of the simulation (see section Modeling Network Formation): not all nodes are influenced by economic size during link formation, but all nodes are supposed to take into account at least node popularity on X. For instance, two institutions, or two donors, can still form a link but this one will be proportional to disease level and social popularity rather than economic size. This asymmetry in link formation for non - receivers' node creates an excess of links formed by popularity reasons.

## Experiment 2

In the second experiment, we vary the ratio $\lambda$ of popularity and economic size together with the incidence of the disease for each node: for every link we compute the average of the two levels (popularity and economic size) to obtain a unique measure $P$ for each candidate link to be compared in Eq 2 with a random uniform probability level $R$. Specifically, we consider that popularity and economic size (independent variables) directly influence the number of edges (dependent variable) and thus the sigmoid shape of the curves while the incidence of disease impacts only on the relation between the ratio $\lambda$ (popularity and economic size) and the number of formed links by provoking a moderate translation of the curves (Fig 3).

In detail, during the process of the link formation, the incidence disease level is changed linearly in an arbitrary multiplicative interval (0,2) for all nodes (from 0% to 200% change in disease incidence). This interval of values is large enough to capture the link formation in the two regimes (a,b) when the disease incidence is low or high. It is noteworthy that in real terms an increase of 200% of the disease level for a consolidated and diffused disease such as TBC is a high value. The result of this simulation is shown in Fig 3.

Due to the linear trend of the disease level, although the effect of the disease incidence is intense, only a small shift (upward and downward) of the curves of link formation (the curves will move upward when the incidence is higher, and on the contrary, downward when incidence is lower) is shown. We interpret this small change in the link formation by assessing that incidence levels can vary only in limited intervals (we don't expect for a specific disease such as TBC an incidence that becomes 100, or 10000 times larger than before, this in general happens in the case of pandemic) while, due to the exponential nature of both X popularity and economic size statistics, we expect that this mechanism of link formation becomes much more relevant. The exponential distribution of popularity and economic size will prevail over the linear changes of the disease incidence and this suggests that underlying economic/marketing or popularity processes should, as in the classical Pareto 20-80 rule, assign a large size to few nodes and a very small size to the large majority of them.

## Experiment 3

The final experiment, shown in Fig 4, studies the effects of changing the fraction of donors (as exogenous variable) in a range between 5% and 20%. We obtain that, if during link formation process, there are many more donors, in the relation donors-receivers there will be a greater probability to form a link and the effect of having more links is evident (shift upward of the curves relative to the link formation for economic or popularity reasons). The remaining part of the formed links (among pairs that do not contain any donor, i.e. institution - institution, receiver - receiver, etc) instead will remain almost constant (green, red and blue horizontal curve named respectively donors 5% -other,donors 25% -other, donors 50% -other). The overall effect of an increased fraction of donors is reflected in the larger number of links that are formed by popularity/economics reasons.

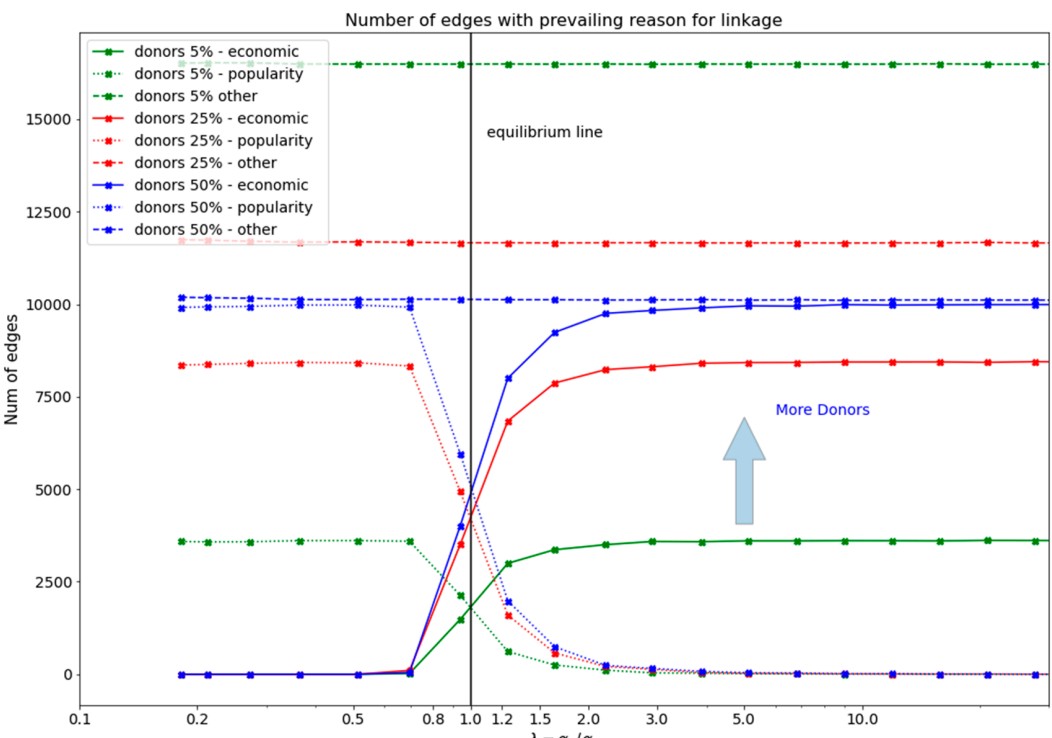

**Fig 4. Changing the fraction of donors there is an evident rise of links formed by economic/popularity interplay.**
Receiver nodes can more easily form a connection if the number of donors is higher.

## Discussion

In Table 1 we reported the values at 10th and 90th quantile relative to the two distributions: popularity on X (formerly Twitter) and economic size from the d-portal database. This interval (Q10-Q90) is larger for the economic size and smaller for the popularity and corresponds to different $\alpha_e$ and $\alpha_p$ indicating the prevalence, within experimental data, of the economic dispersion. This in turn suggests that the effect on the node formation is likely driven by economic interest: receivers in forming a connection will look more for economically powerful nodes (large economic size) rather than their popularity. Crucially, our network model works precisely because of the "different scales" of popularity and economic size. This distinction allows us to demonstrate that economic size is the primary driver for connections in the TBC X social network.For NGOs nodes we assumed the "economic size" relevant in terms of potential funding opportunities. This economic interest is directly reflected into how nodes form connections within the network model. Popularity, instead, plays a significant role only for entities of similar economic size and similar operational roles. In fact, our analysis indicates that popularity only becomes the dominant factor when the economic interest is weak. The network model's logic is designed to capture this dynamic when a link is formed. In summary, it is the different scale of these two variables (popularity and economic size) that explains the link formation in the original network. Finally, we notice that on a platform like X (formerly Twitter), economic size is not explicitly attached to user nodes. A significant aspect of our contribution is filling this data gap: prior to our work with d-portal in collecting financial exchanges among NGOs, there was no accessible information regarding the characteristic economic size or distribution of these organizations. This also brings

**Table 1. Network sizes (nodes and links) and popularity/economic size at 10 and 90th quantiles are used for the X popularity while from d-portal has been inferred the node economic size. The corresponding fitted Exponential coefficient $\alpha$ shows that - according to the distribution of the nodes' sizes - it is the economic size that prevails in comparison with the X popularity in the link formation of the network. The two networks (X popularity and economic size) can have a different level of overlap sitting in the interval 20–40%. To discover common nodes in both networks it is used a natural language processing algorithm that assesses the similarity of node names by computing string distance [21]. The level of string similarity necessary in order to consider two nodes coincident can be selected as high or low. In one case, the overlap will be higher (more nodes will coincide), while in the other (high similarity needed) less nodes will be seen as equivalent. For these reasons, the d-portal (for economic size)-X (for popularity) node overlap can vary in the interval 20–40% if we assume a strict rule (higher overlap) or a mild rule for similarity (lower overlap).**

| Platform | Measure | Type | Value |
|---|---|---|---|
| X | number of nodes | | 1838 |
| X | number of edges | | 7441 |
| X | q10 | Quantile popularity | 58 |
| X | q90 | Quantile popularity | 8211 |
| X | $\alpha$ | Exp coeff on popularity | 0.0000761 |
| d-portal | number of nodes | # | 2447 |
| d-portal | number of edges | # | 34016 |
| d-portal | q10 | Economic size USD quantile | 36010 |
| d-portal | q90 | Economic size USD quantile | 43359850 |
| d-portal | $\alpha$ | Exp coeff on economic size | 0.0000000429 |
| X, d-portal | node overlap of the networks | | 20 - 40% |

into question the prevailing hypothesis that non-profit organizations and their economic size necessarily scales according to a power law or fat tail.

To validate our model we compared the topology of the reconstructed network with that of the original network [5]. Fig 2B) displays a quantile-quantile plot of these two node degree distributions, demonstrating that, at least when the network formation is simulated using $\alpha_e, \alpha_p$ values from the real networks as in Table 1, we successfully reconstruct a suitable degree distribution.

However, while the reconstructed degree distribution aligns well with real data (Fig 2B), this only confirms that *least one* combination of economic size, popularity, and disease level can reproduce the distribution of the real data. We cannot exclude the possibility of many other combinations nor can we definitively state that our model's dynamics are unique. Indeed, other mechanisms and different network formation models might yield the same degree distribution, making this a necessary but not sufficient condition for assessing the uniqueness of the link formation dynamics. For example, if nodes form cartels or organizations connect via legal contracts, their behavior might follow rules our current model cannot yet replicate. To enhance our analysis beyond simple degree distribution, we've augmented the initial degree distribution analysis with the local clustering coefficient. This node-level metric quantifies a node's propensity to form triangles, indicating the presence of strong social ties. It offers a more in-depth understanding than a simple node degree distribution by exploring the connectivity among a node's immediate neighbors. The simulated distribution of local clustering is similar to the actual one (Fig 2C in ordinary Probability Density Function plot), successfully reconstructing the correct density. Despite this success, the true network contains additional information our model can't fully capture. For example, nodes representing the UN agencies operate internationally and likely exhibit different behavior compared to institutions specifically focused on health, such as the CDC or WHO. The discrepancies observed between the simulated and actual network clustering are likely due to the existence of internal, national, regional, or even role-based network communities. It is important to

notice that our model is intentionally conservative. It focuses solely on the interplay between donors, receivers, and disease levels, deliberately omitting information related to geography, communities, or specific node roles (beyond the primary classification of donors, receivers, institutions, and others).

Another limitation is that, due to the nature of the data drawn from X (formerly Twitter), we cannot reconstruct - as also stated in the our starting paper of [5] - the true temporal dynamics of the link formation, as this information is not available on X.

Nevertheless, the network formation model introduced here, which analyze link creation based on economics and popularity factors, proves particularly helpful in understanding how connections are realized. it suggests that the X friendship network—despite being a social media network—is likely formed with the intent of establishing partnerships aimed at securing more funding, enhancing aid attractiveness, and increasing visibility. our findings indicate that charities and NGOs utilize social network connections primarily for marketing and funding initiatives, leading to a specialized "friendship" network. Finally, as demonstrated, the mechanism of link formation is tied to node sizes, and particularly influenced by the dispersion of node sizes, resulting in a simulated network that accurately captures the node degree distribution. From this dynamic, we can derive useful insights that can inform more effective health policy interventions, especially those targeting large-sized nodes.Overall, the network formation model introduced here to analyze the network formation process between economics and popularity reasons results particularly helpful in investigating the link creation and assessing that the X friendship network - even if it is a social media network - is likely formed with the intent of establishing partnerships targeted to obtain more funding, aid attractiveness and more visibility. In other words, in our opinion, the choice of a social network connection for charities and NGOs is clearly oriented to marketing and funding initiatives and leads to a *specialized* friendship network. Finally, as we have shown, the mechanism of link formation is related to the sizes of nodes and in particular it strongly depends on the nodes' size dispersion, resulting in a simulated network that captures the distribution of node degree. From this dynamic, we can derive useful insights that can help to orient more effective health policy interventions based especially on the big size nodes.

## Conclusion and policy implications

This study aims to explain the formation of the X friendship network specialized in the fight against the TBC by employing a network formation model built according to node popularity based on social network data of X, the economic size drawn from d-portal and the disease incidence (arbitrarily imposed to nodes by normalizing the values of incidence per country drawn from WHO database). We found that, when economic size is distributed with a more skewed shape with respect to popularity and consequently the nodes' size dispersion is larger, we can conclude that it is the economic interest that prevails in the link formation of our network over the popularity of nodes (Fig 2A).

On the other side, the effect of an increased incidence of the disease is slightly reflected in the velocity of the transition from a network *social media oriented* to a network *economic oriented*. In this case, what happens is that, at high-incidence disease levels, the NGOs start earlier to prefer economic reasons when forming a connection. However, the added shift in the number of connections formed by popularity or economics change is low, indicating only a mild effect due to the incidence disease variation (Fig 3). The economic size diversity, combined with a higher fraction of donors is more effective in accelerating the network formation of NGOs. More donors act by improving the fundraising-oriented strategies while favouring more connections with a prevailing economic value (Fig 4): an interesting finding

that may help especially the NGOs operating in low-income countries where popularity oriented strategies at link formation are not particularly rewarding, especially if pursued with only local actors.

According to these findings, we can derive several interesting policy recommendations and healthcare best practices suggestions to improve the actions of Global Health Networks involved in fighting a specific disease such as TBC. The GHNs should try to attract not only big players (highly central in the network, and with large node size) but also those organizations that, although having small size, are instead needing of a more regular cash flow, and may contribute to maintaining a stable and functional network with a better allocation of resources. These insights come from the argument that networks that operates exclusively in developing countries will tend to have two possible structures: a) only international institutions such as UN agencies are operative (this is the case of really poor countries, or during war scenarios when local institutions, and the economic systems are collapsed), b) a large number of powerless, small and underfunded local charities are operative in the territory with no local big players acting as donors. As a result in these countries the distribution of the node sizes is not much skewed and the resulting action-networks can lack of aid-funding attractiveness if all nodes share the same popularity level or the same (low) economic power.

The network *diversity* is, thus a feature of paramount importance for the performance and an effective overall functioning of partnerships aimed at fund-raising policies. As the economic interest is the key driver for the linkage to improve the network partnership it is necessary that the majority of non-institutional nodes, like donors and receivers, be a large fraction of the network. If the network, instead, is overall made by institutional nodes (official organizations) and/or health-related experts, the network performance (measured by network size, connectivity, centrality etc.) will likely be poor - even if the funding level is high. An example in this sense is the GHN of Pneumonia that in the work of [5] the GHN of Pneumonia is analyzed identifying the most central nodes, exploring its size and density using the standard network analysis indicators (centrality measures). The findings of that study shows the weakness of the Pneumonia network with only 65 nodes and 196 links compared with the more mature, active and global network of TBC, that has 1838 nodes and 7441 links. Specifically, the GHN of Pneumonia - before the COVID-19 pandemic - was poorly connected and lacking global coverage with most of its nodes represented by official organizations such as the WHO, or Stop Pneumonia and having only a smaller fraction of organizations devoted to a specific Pneumonia related cause. In this regard, in the paper of [5] the authors introduce the concept of "Focus" of a network as a measure of how many nodes share in their name a clear indication of the mission (for instance "StopPneumonia" is an account name clearly targeting Pneumonia, while "AfricaWellBeing" is not). According to this finding, Pneumonia resulted in the network with the smallest focus, a prevalence of institutions, and this suggests a strong weakness of this GHN in fighting the disease (see Table 4 in [5] for further details).

In our study relative to the network formation model, we suppose that other than having a limited focus these poor GHN have also strong lack of node diversity, less economic interest and quantitatively fewer donors, and receivers. Finally, these networks do not have enough attractiveness for nodes, as they lack any "skeweness effect" in economic and popularity distribution.

Starting from these points policymakers and governments working with official, transnational health organizations such as the Global Fund or the WHO should then promote a higher diversification of their actors, trying to attract NGOs that despite their small size may help to get more funding initiatives with their specialized action. Network diversification is an important value that promotes network stability and efficiency and in turn improve health equity.

While it is promising, this study has a few limitations: the X platform does not allow to recover the exact moment in time when two users became friends, this implies that the exact sequence of link formation remains unknown. We can suppose that the most powerful organizations, with an active media strategy, would have been "early adopters" of the X technology and had formed friendship connections before others less powerful organizations.

Moreover, it is not obvious to retrieve the correspondence between social networks links and economic partnerships. We believe that the total flow of money that a company has received is a good proxy of its economic size. But it is also possible that some institutions could simply act as "brokers" that distribute funds to others, and despite the elevate flow they are not the"true players" in GHNs partnership. In future research, we plan to extend the network formation model to other domains, extending our sample of NGOs to that fight other infectious diseases and for a better generalization including other leading causes that may impact on the formation of a link such as being part of big initiatives that involve many international actors, introducing competition, and the formation of "cartels" or internal partnerships that aggregate nodes into communities/clusters favouring phenomena of polarization that can modify collaboration creating further friction. We expect to derive useful insights on the way the formation of powerful communities relative to specialized nodes can hinder, or enhance, innovative actions and successful strategies. Overall,the present study examines an interesting area of research (social network formation in the health and funding activities) with a powerful methodology based on a rule-based model of network formation that can help to reconstruct, and test the different mechanisms of link formation. The study represents a relevant starting point for useful suggestions in the health policy debate with the aim of improving the fighting of a specific disease, optimize funding strategies and gain in network resilience.

## Supporting information

**S1 File. The manuscript has an attached "Supplementary information" file reporting tests for the ML analysis of the TBC network classes, and the Confusion Matrix that shows the performance of the ML estimator.**
(PDF)

## Author contributions

**Conceptualization:** Milena Lopreite.

**Formal analysis:** Milena Lopreite, Michelangelo Puliga.

**Investigation:** Mauro Gallegati, Milena Lopreite, Michelangelo Puliga.

**Methodology:** Mauro Gallegati, Milena Lopreite, Michelangelo Puliga.

**Resources:** Michelangelo Puliga.

**Software:** Michelangelo Puliga.

**Supervision:** Milena Lopreite.

**Validation:** Milena Lopreite.

**Visualization:** Milena Lopreite.

**Writing – original draft:** Mauro Gallegati, Milena Lopreite, Michelangelo Puliga.

**Writing – review & editing:** Mauro Gallegati, Milena Lopreite, Michelangelo Puliga.

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
