## [Decision Letter · Decision Letter 0]

10 Jul 2025

PONE-D-25-26747Modeling the formation of a worldwide health network fighting TBC: key drivers in policy, management and governance in developing countries and global health institutionsPLOS ONE

Dear Dr. Lopreite,

Thank you for submitting your manuscript to PLOS ONE. After careful consideration, we feel that it has merit but does not fully meet PLOS ONE’s publication criteria as it currently stands. Therefore, we invite you to submit a revised version of the manuscript that addresses the points raised during the review process.

Dear authors,

the reviewers completed the review process. One of them suggests accept, the other major revision.

However, the requests from the reviewer suggesting major revision are mostly clarifications that I think should be anyway addressed.

Overall, I am suggesting minor revision as I think it is most appropriate for your case.==============================

We look forward to receiving your revised manuscript.

Kind regards,

Matteo Cinelli

Academic Editor

PLOS ONE

2. For studies involving third-party data, we encourage authors to share any data specific to their analyses that they can legally distribute. PLOS recognizes, however, that authors may be using third-party data they do not have the rights to share. When third-party data cannot be publicly shared, authors must provide all information necessary for interested researchers to apply to gain access to the data. (https://journals.plos.org/plosone/s/data-availability#loc-acceptable-data-access-restrictions)

3. Please update your submission to use the PLOS LaTeX template. The template and more information on our requirements for LaTeX submissions can be found at http://journals.plos.org/plosone/s/latex"

Additional Editor Comments:

Dear authors,

the reviewers completed the review process. One of them suggests accept, the other major revision.

However, the requests from the reviewer suggesting major revision are mostly clarifications that I think should be anyway addressed.

Overall, I am suggesting minor revision as I think it is most appropriate for your case.

Reviewers' comments:

Reviewer's Responses to Questions

**Comments to the Author**

1. Is the manuscript technically sound, and do the data support the conclusions?

Reviewer #1: Yes

Reviewer #2: Partly

2. Has the statistical analysis been performed appropriately and rigorously? 

Reviewer #1: Yes

Reviewer #2: No

3. Have the authors made all data underlying the findings in their manuscript fully available?

Reviewer #1: Yes

Reviewer #2: Yes

4. Is the manuscript presented in an intelligible fashion and written in standard English?

Reviewer #1: Yes

Reviewer #2: Yes

5. Review Comments to the Author

Reviewer #1: I enjoyed reading the authors' article and found their motivations and research design to be interesting and compelling.

The research is technically sound and employs appropriate statistical and network tools. The choice of integrating the network of financial flows with the social network interactions in X is very compelling in modeling the interactions of actors that partake in a health network.

The experiments are explained clearly and are performed rigorously. The results are both illustrated in the figures and explained in captions and in the main text of the article, making them accessible to the reader.

Reviewer #2: In the manuscript, the authors introduce a network formation model to identify the drivers of the health network in the fight against TBC. Further, they investigate how social media popularity, economic size, and disease incidence affect network growth.

I enjoyed reading the study and found their contribution to be interesting. However, I believe the manuscript could benefit from further clarification regarding the proposed methodology, as well as additional experimental details. Below, I outline my main points of concern.

ML model for user labeling

In Introduction and Methods, the authors mention that, starting from an X friendship network collected in prior work, they employ a Random Forest classifier to categorize users into four classes (institutions, donors, receivers, others), achieving an 85% accuracy. However, some important details are missing from the presentation.

1. What features was the model trained on? Were there any pre-processing steps involved?

2. How balanced are the classes? Unless classes are very balanced, the F1 score would be a more representative metric. Even better, the authors could display a confusion matrix that compares true and predicted classes.

3. Is the reported performance the average across CV samples (how many?) or on a separate test set? In the latter case, how large (in terms of number of accounts) is the training/validation set vs the test set?

Network formation model

1. The authors mention that the number of followers and “economic size” are both characterized by skewed distributions. This is reasonable. However, plotting the two distributions with a fit line or a goodness-of-fit test would better motivate the use of the exponential distribution.

2. Given that popularity and economic size are on different scales (see the quantiles in Table 1), won’t the average popularity be typically smaller than the average economic size (Eq (2) in SI)? Is this something that may affect the results, or have the authors taken it into account? In the Discussion, the authors say that node formation is likely driven by economic size, but unless some form of standardization is applied, this may just be a matter of scale.

3. I think a clearer definition of the probability P (Eq (2) in SI) should be provided.

4. I’m not confident that comparing the degree distributions is the better way to validate the simulation outcomes. The authors themselves acknowledge that “different network formation models can eventually reproduce the same degree distribution”. For instance, as an additional metric, the authors could compute the Jaccard similarity of the two edge sets.

5. In Fig. 1, I don’t understand why two curves (economics prevailing or popularity prevailing) are shown instead of just one. Isn’t it just the value of lambda that determines which of the two is prevailing?

Minor points and suggestions

- the manuscript’s writing could be improved, and some sentences should be streamlined to enhance the paper’s overall readability and avoid ambiguities

- the Results section could be split into subsections dedicated to each experiment

- as they are the backbone of the contribution of this work, I think Eq (1) and Eq (2) in the Supplementary Information should be moved to the main text along with their explanation

6. PLOS authors have the option to publish the peer review history of their article (what does this mean?). If published, this will include your full peer review and any attached files.

Reviewer #1: No

Reviewer #2: No

---

## [Author Response · Author response to Decision Letter 1]

31 Jul 2025

All our comments and response to the reviewers are in the attached referee's report.

---

## [Editor Report · Decision Letter 1]

4 Aug 2025

Modeling the formation of a worldwide health network fighting TBC: key drivers in policy, management and governance in developing countries and global health institutions

PONE-D-25-26747R1

Dear Dr. Lopreite

We’re pleased to inform you that your manuscript has been judged scientifically suitable for publication and will be formally accepted for publication once it meets all outstanding technical requirements.

Kind regards,

Matteo Cinelli

Academic Editor

PLOS ONE

---

## [Editor Report · Acceptance letter]

PONE-D-25-26747R1

PLOS ONE

Dear Dr. Lopreite,

I'm pleased to inform you that your manuscript has been deemed suitable for publication in PLOS ONE. Congratulations! Your manuscript is now being handed over to our production team.

Kind regards,

on behalf of

Dr. Matteo Cinelli

Academic Editor

PLOS ONE